# Clinical Manifestation and Diagnostic Process of Celiac Disease in Poland—Comparison of Pediatric and Adult Patients in Retrospective Study

**DOI:** 10.3390/nu14030491

**Published:** 2022-01-23

**Authors:** Emilia Majsiak, Magdalena Choina, Alastair M. Gray, Mariusz Wysokiński, Bożena Cukrowska

**Affiliations:** 1Department of Basic Nursing, Chair of Development in Nursing, Faculty of Health Promotion, Medical University, 20-081 Lublin, Poland; 2Polish-Ukrainian Foundation of Medicine Development, 20-701 Lublin, Poland; office@pufrm.eu; 3Health Economics Research Centre, Nuffield Department of Population Health, University of Oxford, Oxford OX3 7LF, UK; alastair.gray@dph.ox.ac.uk; 4Department of Basic Nursing, Chair of Development in Nursing, Faculty of Health Sciences, Medical University, 20-081 Lublin, Poland; mariusz.wysokinski@umlub.pl; 5Department of Pathomorphology, The Children’s Memorial Health Institute, 04-730 Warsaw, Poland; b.cukrowska@ipczd.pl

**Keywords:** celiac disease, diagnostic process, gluten free diet, delayed diagnosis

## Abstract

The diagnosis of celiac disease (CD) may be delayed due to non-specific clinical symptoms. The aim of the study was to evaluate the clinical manifestation and diagnostic process of CD in Polish children and adults. Methods: The members of the Polish Coeliac Society (*n* = 2500) were asked to complete a questionnaire on socio-demographic factors, clinical and diagnostic aspects of CD. The analysis was based on 796 responses from patients with confirmed CD diagnosis, and included 224 (28.1%) children and 572 (71.9%) adults. Results: The mean duration of symptoms prior to CD diagnosis in children was significantly shorter than in adults (*p* < 0.001), and amounted to 3.1 and 9 years respectively. The most frequent symptoms before CD diagnosis were abdominal pain and bloating in children (70.4%), and chronic fatigue in adults (74.5%). Although almost all CD patients claimed to strictly avoid gluten after CD diagnosis, symptoms were still present in the majority of these respondents. No comorbid diseases were reported by 29.8% of children and by 11.7% of adults (*p* < 0.001). Conclusions: the results indicate that CD diagnosis is delayed in Poland, especially in adults, and clinicians should be aware of the diversity in CD presentation.

## 1. Introduction

Celiac disease (CD) is regarded as one of the most common autoimmune disorders of the alimentary tract with a global prevalence based on serologic test results of 1.4%, with evidence of increasing prevalence over recent decades [1]. Research indicates that the incidence of CD is highest in women and children [2].

The “classical” type of CD, which occurs mainly in younger paediatric patients, is characterized by chronic diarrhea and malabsorptive features [3]. In the non-classical CD, which is more and more often observed in both children and adults, symptoms from outside the gastrointestinal tract, such as short stature, anemia, decreased bone density, skin changes, chronic fatigue, headaches and depression predominate [4]. The varied clinical presentation of the disease may contribute to a significant delay or even failure to diagnose CD [5]. Early detection of CD and the implementation of gluten-free diet (GFD) may improve the quality of life of CD patients and reduce the costs of diagnostic processes and subsequent therapy [6]. Undiagnosed CD could result in serious complications, including oncological diseases and infertility [7]. A delay in CD diagnosis may affect not only somatic, but also psychological well-being. Symptomatic CD patients who are undiagnosed are likely to take medicines or use health care services excessively. An increase in days of sickness and doctor visits as well as in intake of antidepressants and drugs for dyspepsia has been observed among Finnish patients with delayed CD diagnosis [4,8].

In our recent study, we reported that the time from first symptoms to diagnosis of CD in Poland lasted 7.3 years on average. Although long, this was significantly shorter (*p* < 0.001) than the comparable CD diagnostic delay in the United Kingdom of 13.2 years [9]. This difference might be partly explained by the fact that there were more children in the Polish study group than in the British study group (28.1% and 12.4%, respectively). As children constituted almost one third of the Polish study group, the duration of symptoms prior to CD diagnosis in Polish children and adults should be verified. Thus, the aim of the current study was to identify and compare differences in the diagnostic process of CD among paediatric and adult CD patients.

## 2. Materials and Methods

### 2.1. Study Design

The study was performed among members of the Polish Coeliac Society (www.celiakia.pl (accessed on 20 December 2021)). This society includes patients with diagnosed CD and those who adopted a GFD on their own. Two and a half thousand members of the society were sent a letter with information on the planned study and a request to complete the attached questionnaire. Of this number, 969 (38.76%) surveyees returned the questionnaire. Patients who had adopted a GFD individually without a CD diagnosis (*n* = 173) were excluded from the study, leaving 796 questionnaires included in the analysis. Parents or guardians were asked to complete the questionnaire on behalf of children (under the age of 18). The study was conducted with the consent of the Bioethics Committee of the Children’s Memorial Health Institute (No. 48/KBE/2017).

### 2.2. Questionnaires

The detailed description of the questionnaire was presented in our recently published paper [9]. The questionnaire was based on the original questionnaire, “The Impact of Coeliac Disease on Your Life: A Survey of Your Views”, developed by Gray and Papanicolas with the consent of the authors [10]. Respondents were asked to answer questions about socio-demographic factors, diagnostic and clinical aspects and CD-related costs they had incurred. The questionnaire distributed among members of Polish Coeliac Society was modified compared to the original British questionnaire. In particular, a question concerning the decision to start GFD was added, to allow respondents who had adopted a GFD without a CD diagnosis to be excluded. Surveyees who indicated that a GFD had been adopted after a CD diagnosis were then asked about the method in which the diagnosis was made: blood test (without specification of the serological tests), duodenal biopsy, genetic examination or any other type of examination, which they were asked to specify. Surveyees were also asked to indicate the specialization of the clinician who had made the CD diagnosis. 

### 2.3. Statistical Analysis

Statistical analysis was performed using Statistica 10 (StatSoft Poland, Kraków, Poland). Two variables were compared using Wilcoxon’s test, while three variables were compared using the Kruskal–Wallis test. Due to the fact that the Kruskal–Wallis test indicates only the presence of statistically significant difference, Bonferroni’s test was performed for further analysis. Chi-squared tests were used to define statistically significant correlations between qualitative variables. To test the statistical significance of differences by gender, the Fisher exact test was used. To measure correlation between variables, Spearman’s rank correlation coefficient was used. Continuous variables were summarized using mean values, while variability around mean values was reported in terms of standard deviations (SD). The precision around mean values was described with 95% confidence intervals (CI). A *p*-value < 0.05 was considered as statistically significant.

## 3. Results

### 3.1. Characteristics of Paediatric and Adult Patients 

A total of 796 questionnaires were returned, of which 224 (28.14%) referred to children and 572 (71.86%) adults (Table 1). A majority of respondents were female in both groups—58.5% (131/224) among children, and 89.3% (511/572) among adults, respectively. The average age of patients at CD diagnosis was 6.8 years in the paediatric group and 34.3 years in the adult group, while mean age at survey was 9.7 years (range 2 to 18) and 37.2 years (range > 18 to 80) in children and adults, respectively. The longest reported duration of diagnosed CD was 53 years.

### 3.2. Clinical Symptoms

Respondents were asked to report the symptoms they had experienced and their mean duration before CD diagnosis (Table 2). Abdominal pain/bloating (69.3%), chronic fatigue (61%), flatulence (58.4%), anemia (55.2%) and diarrhea (54.5%) were the most common symptoms in the whole analysed group, regardless of age. Adults stated that the most frequent symptoms were: chronic fatigue (74.5%), abdominal pain/bloating (68.7%), anemia (63.7%) and flatulence (62.4%). The most common symptoms among children were: abdominal pain/bloating (70.4%), flatulence (51.7%), diarrhea (48.6%) and anemia (40.8%). Most of the reported symptom types (excluding abdominal pain/bloating, skin rash and weight loss) were significantly more common in adults than in children prior to CD diagnosis, notably: diarrhea (*p* = 0.011), constipation (*p* = 0.008), osteoporosis (*p* = 0.005), flatulence (*p* = 0.003), chronic fatigue, headaches, mouth ulcers, anemia and depression (*p* < 0.001). No symptoms prior to CD diagnosis were declared by 4% of children and by 3% of adults (Table 2).

### 3.3. Duration of Symptoms before CD Diagnosis

The mean duration of symptoms before CD diagnosis varied substantially by symptom, ranging from 4.7 years for diarrhea to 9.2 years for anemia (Table 2). The mean duration of any symptom before establishing the diagnosis was 7.3 years for the whole analyzed group. Symptoms lasted on average significantly longer in adults than in children, at 9.0 and 3.1 years respectively (*p* < 0.001). The symptom experienced for the longest time before CD diagnosis in adults was anemia (11.6 years on average), whereas in children it was depression (4.1 years on average), although it should be noted that only 5.8% of respondents in the pediatric group reported depression. Among adults, the symptom that lasted for the shortest period prior to CD diagnosis was diarrhea (5.8 years on average), and in children osteoporosis (0.6 years on average).

### 3.4. The Effect of GFD on CD Symptoms

The respondents were asked about GFD adherence after CD diagnosis. Almost all children (98.7%) claimed to follow GFD all of the time. The percentage of adults who reported adhering to a GFD at all times was significantly lower in comparison with children (91.9%, *p* < 0.001). Adherence to GFD most of the time was reported by 46 respondents (5.7%): 2 children (0.9%) and 44 (7.7%) adults (*p* < 0.001). One child admitted to not following a GFD, and two adults reported eating gluten-free products rarely.

In spite of CD diagnosis and subsequent adherence to a GFD, many respondents reported persisting symptoms of CD (Table 3). The most frequent symptoms after CD diagnosis among all patients were: flatulence (41.8%), abdominal pain (39.0%), chronic fatigue (34.9%) and headaches (28.2%). No symptoms were reported by 155 (19.4%) of all CD patients, a percentage that was significantly higher in children than in adults (29.3% versus 15.6%, *p* < 0.001). Among children, the most common symptoms which persisted despite GFD diet were: abdominal pain (37.0%), flatulence (31.6%), headaches (20.0%), chronic fatigue (19.3%) and skin rash (19.3%). The most frequent symptoms among adults after CD diagnosis were: flatulence (47.8%), chronic fatigue (44.0%), abdominal pain (40.2%) and headaches (33.0%). The symptom that lasted for the longest time in children after diagnosis was headaches (5.5 years), while in adults it was weight loss (6.4 years). The mean duration of symptoms after diagnosis in both age group was similar, except for anemia (1.4 years in children and 3.6 years in adults, *p* = 0.001).

### 3.5. Comorbidity

The coexistence of comorbid conditions known to be associated with CD was also assessed (Table 4). Thyroid diseases (23.4%), depression (12.6%), immunoglobulin A (IgA) deficiency (10.7%) and miscarriages (9.4%) were the most frequently reported comorbid disorders in the whole study group. Adults reported slightly but significantly higher (*p* < 0.001) occurrence of comorbid medical conditions (88.3%) compared to the pediatric group (70.2%). The most common comorbid diseases in adults were thyroid diseases (28.8%), which occurred almost three times more often than in children (9.8%, *p* < 0.001). In children, IgA deficiency was the most common comorbidity (13.3%), and there was no significant difference in the occurrence of IgA deficiency between children and adults (13.3% versus 9.6%). Many adult patients (42.7%) reported having other diseases than those specified in the questionnaire, such as epilepsy, peptic ulcer disease, colitis ulcerosa, Crohn’s disease, irritable bowel syndrome, ischemic heart disease, hypertension, gastroesophageal reflux, psoriasis, endometriosis, albinism and allergic diseases (food and pollen allergy, atopic dermatitis, bronchial asthma). Children (34.2%) also reported other diseases than those specified in the questionnaire, such as gastroesophageal reflux, peptic ulcer disease and allergy (atopic dermatitis, food and pollen allergy, bronchial asthma).

### 3.6. Diagnostic Process

Almost all children had undergone serological tests (*n* = 270, 91.8% of all children included to the study). Out of all children, 67 (22.8%) had been diagnosed without duodenal biopsy: 23 children before the year 2012 and 44 children after the year 2012. HLA-typing had been performed in 117 (39.7%) children and in 94 (18.7%, *p* = 0.0001) adults (Table 5). Among adults, most (83.2%) had undergone duodenal biopsy, and serological tests had been performed at significantly lower frequency than in children: 79.0% and 91.8% respectively (*p* = 0.0001).

We also asked questions about the number of appointments with General Practitioners (GPs) due to symptoms which occurred prior to CD diagnosis, and this information was provided by 792 (99.5%) respondents. The average number of appointments with GPs was 17.7. Analysis using the Spearman’s rank correlation coefficient showed a statistically significant relationship between the number of visits and the mean duration of all symptoms (*p* < 0.001). The longer the symptoms lasted, the greater the number of visits (Table 6). There was no statistically significant difference between children and adults regarding the number of appointments with GPs due to symptoms prior to CD diagnosis.

## 4. Discussion

Previous studies have suggested that an increasing proportion of patients diagnosed with CD lack prior gastrointestinal symptoms, which have been considered a feature of classical CD [3,11]. In spite of increasing prevalence of nonclassical CD manifestation, our analysis showed that gastrointestinal symptoms remain the most frequent CD manifestation among Polish patients [12]. Abdominal pain, flatulence and diarrhea were the most common CD symptoms among children, and these results are in line with an American analysis reporting that 82% of children diagnosed with CD had a gastrointestinal manifestation [13]. We found that in adults the clinical picture was more varied than in children, and although gastrointestinal symptoms were very common (flatulence and abdominal pain were observed in 62% and 68%, respectively), fatigue was the most common symptom (75%). Anemia was also one of the most commonly reported symptoms in both children and adults. Interestingly, some gastrointestinal symptoms such as diarrhea, constipation, and flatulence were significantly more often present in adults than in children. The most common extraintestinal symptoms reported by Polish children in our study were chronic fatigue and skin rash. In comparison, Jericho et al. found that short stature (33%), fatigue (28%), and headaches (20%) were the most common extraintestinal symptoms in CD children. The same authors have shown that iron deficiency anemia (48%), fatigue (37%), and headaches/psychiatric disorders (24%) were common in adults [12].

Undoubtedly, heterogenous clinical manifestation of CD contributes to delays in diagnosis. As CD symptoms are not specific, they remain unrecognized as CD and represent a challenge for clinicians [4].

Regardless of age, Polish patients waited slightly longer than 7 years on average for a CD diagnosis to be established. The symptom with the longest duration prior to CD diagnosis was anemia (more than 9 years), but other symptoms reported as lasting more than 7 years included headache, constipation, flatulence, joint pains, skin rash, mouth ulcer and bloating (8.6; 8.5; 8.1; 7.6; 7.4; 7.2 and 7.1 years, respectively). Our analysis showed that adults waited significantly longer for the proper diagnosis than children (9 years versus 3.1 years). Similar results were reported by Vavricka et al., regarding CD patients in Switzerland [11]. They found that total diagnostic delay was significantly higher in patients over 30 years of age at diagnosis. In other countries, the delay in CD diagnosis ranged from 9 to 13 years [4], but it is worth mentioning that these studies were performed in the years 2001–2012, so the time shift between them and our study could affect the results.

Our results show that delayed diagnosis of CD remains a problem in Poland, especially in adult patients. We found that duodenal biopsy was more common among adults (83.2%) than children (77.8%), while serological tests were performed in fewer than 80% of adults but over 90% of children. These differences result from the current diagnostic recommendations for CD. According the ESPGHAN guidelines, CD in children starting from 2012 can be recognized without intestinal biopsy [14,15], whereas in adults the histopathological analysis of duodenal specimens is still a gold standard [14,16]. We suggest that diagnostic delays in adults can be associated with a non-specific varied clinical picture, but also with this diagnostic approach based on histopathological assessment. A recent study performed in the United Kingdom revealed that the majority of endoscopists did not follow guidelines for diagnostic endoscopy for CD, which led to a reduction in diagnosis rates by over 50% [17]. In addition, the longer diagnostic delay observed in adults might result from the fact that, in Poland, CD has been regarded as a pediatric disease until quite recently [9]. The Polish manual “Internal diseases: a manual for medical students”, edited by Franciszek Kokot and issued in 1991, states that “Since celiac disease is a pediatric disorder, it will not be presented in detail in this manual” [18]. The description of CD in the 6th issue of the same book, published in 1996, was not expanded: “(CD) disease will not be presented in this book since it is a pediatric disorder” [19]. This signifies that in Poland in the 1990s, CD was still considered as a pediatric disease, which would be consistent with the pattern of diagnosis reported in this study.

It has been proven that even a short delay in CD diagnosis results in an increased health burden at the individual as well as at society level [4]. Our analysis revealed that the number of symptom-related appointments with GPs made by celiac patients increased in line with the duration of these symptoms. We also found a correlation between the duration of the diagnostic process, the number of coexistent medical conditions and the duration of CD symptoms prior to diagnosis. Results from a Finnish study were similar to ours: patients who experienced a delay in CD diagnosis of longer than 3 years made more primary health care visits and reported more days of sickness [4] than those diagnosed more rapidly. The Finnish authors also reported an increased use of antidepressants, analgesics and medicines for dyspepsia in the delayed diagnosis group, an analysis which our study did not perform.

At present, a GFD is the only known and effective treatment for CD [20]. We showed that although almost all CD patients reported strict adherence to a GFD, they still had CD symptoms that lasted for 2.6 years on average after the diagnosis. It was observed that among celiac pediatric patients, symptoms receded earlier after eliminating gluten than in celiac adults, regarding almost all analyzed symptoms, except for headaches, joint pain, depression and ataxia. Nevertheless, a statistically significant difference in duration of symptoms after CD diagnosis was observed only in the case of anemia.

Strict adherence to GFD imposes numerous restrictions on the patient, having both financial and social implications [21]. What is more, the complete elimination of gluten from the diet does not guarantee complete mucosal recovery [22]. What could also contribute to persistent inflammation in bowel mucosa is the cross-contamination of gluten-free products with wheat, barley and rye [21]. This might explain why CD symptoms are still present in patients who report adherence to GFD. The levels of antibodies against tissue transglutaminase type 2, both in children and in adults, may normalize a year or more after the introduction of GFD [23], during which time the disease is still active.

Comorbidity in CD has been widely discussed, and correlations have been reported between the prevalence of CD and some genetic disorders such as Down syndrome, Turner syndrome, Williams syndrome and IgA deficiency [24]. In Polish CD patients, we found that comorbidities were significantly more frequent in adults than in children (respectively, more than 88% in comparison to 70%, *p* < 0.001). A meta-analysis performed in 2016 showed that the prevalence of thyroid diseases in children with CD was higher than in adults with CD (more than 6% compared to less than 3%) [16]. In contrast to these findings, we found that thyroid diseases were the most common coexistent disease in Polish CD adults (almost 30%) but less prevalent in Polish pediatric CD patients (almost 10%). However, a strong correlation between CD and autoimmune thyroid diseases is proven and ESPGHAN Guidelines recommend performing serological screening for CD in children with thyroid diseases [14]. The most common concurrent disease among Polish children with CD was IgA deficiency, affecting more than 13% of underage patients. The frequent coexistence of IgA deficiency and CD (about 10% of CD patients) explains why serological screening of patients should be started with determination of IgA against tissue transglutaminase type 2 and total IgA [25]. ESPGHAN Guidelines also suggest screening children with type 1 diabetes mellitus for CD, as they are known to be at greater risk for CD [14]. The common coexistence of CD and type 1 diabetes mellitus in Polish children (more than 5%) was also observed.

Patients with CD, compared to healthy individuals, more frequently demonstrate concurrent autoimmune diseases (about 5%) [26]. One possible explanation of this phenomenon is the fact that some autoimmune disorders share a genetic background [27]. Another factor that may promote the development of other autoimmune diseases in CD patients is the time of exposure to gluten. In 1999, Ventura et al. revealed that delayed CD diagnosis, and consequently longer exposure to gluten, was associated with higher prevalence of autoimmune diseases [28]. The role of gluten in the development of autoimmunity was also shown in a study performed by Cosnes et al. [29]. However, some authors have found no association between delayed CD diagnosis, longer exposure to gluten and development of autoimmune disorders [30]. Our analysis revealed that there was an association between age at CD diagnosis (*p* < 0.001), the number of comorbid autoimmune disorders (*p* < 0.001) and the mean duration of CD symptoms prior to CD diagnosis.

A delayed diagnosis can also lead to the development of secondary autoimmune diseases such as neurological diseases, which are marked by the occurrence of anti-neuronal and anti-ganglioside autoantibodies [31]. Moreover, a lack of delayed diagnosis and GFD may affect the incidence of osteopenia/osteoporosis as has been demonstrated recently [32]. We were particularly surprised by the number of reported food and inhalation allergies in our patients with CD. There is a question whether delayed GFD introduction could be connected with the induction of an allergy or be a novel allergy. Therefore, a more detailed analysis of comorbidities in the studied patients with CD is needed, which will be the subject of further considerations.

## 5. Limitations of the Study

We are conscious of the limitations of the study. First, a response bias cannot be excluded due to the retrospective character of the assessment. However, there are no large long-term prospective studies with frequent data collection and the retrospective method has been widely used in studies on CD and on other diseases [33,34,35]. Our analysis is based on data concerning tests and symptoms and their duration as reported by patients, not on medical notes or records, and so there may be errors of recall or understanding. We were unable to compare our results with medical data on specific antibody serological tests and histological tests, but we asked in the questionnaire which criteria were used for CD diagnosis and by whom CD was diagnosed. The percentage of respondents who indicated a diagnostic approach other than ‘biopsy’/‘biopsy plus blood test’/‘blood test’, were only 1.9% (15/796), and in this small group, the percentage of respondents who indicated that a diagnosis was not made by a gastroenterologist was only 0.75%. Therefore the likelihood that our results might be biased by diagnoses that were not truly medically confirmed must be low. A final limitation is that we cannot exclude the possibility that the respondents are somehow unrepresentative of the entire group of Polish patients with CD. We cannot therefore guarantee that the outcomes of an analogous prospective study would be similar, but there are at present limited data on CD epidemiology in Poland to compare with our results, and we would welcome further studies in this area.

## 6. Conclusions

Our study indicates that the process of diagnosing CD in Poland is still subject to significant delays. Polish patients typically experience more than 7 years of symptoms (9 years for adults, 3.1 years for children) before a CD diagnosis is established, and during this time they frequently consult GPs about their symptoms. After diagnosis, almost all CD patients claimed to be strictly avoiding gluten, but symptoms were still present in the majority of the respondents, with significantly higher frequency in adults than in children. Our results suggest that better awareness among clinicians of the diversity in CD presentation could help to improve the diagnostic process, especially in adults, and that more could be done to help those diagnosed with CD to adhere to a fully GFD.

## Figures and Tables

**Table 1 nutrients-14-00491-t001:** Characteristics of the study group.

Variables		All Patients	Children	Adults
No. of included questionnaires		796	224 (28.1%)	572 (71.9%)
Sex—No. (%)	Female	642 (80.7%)	131 (58.5%)	511 (89.3%)
	Male	127 (19.3%)	93 (41.5%)	61 (10.7%) ^1^
Average age at survey in years—Mean (SD)		29.4 (16.0)	9.7 (3.8)	37.2 (11.7)
Average age at diagnosis in years—Mean (SD)		24.1 (15.9)	6.8 (4.2)	34.3 (10.6)

^1^ There was a statistically significantly higher proportion of females in adults compared with pediatric patients (*p* = 0.0001). Fisher exact test was used for statistical analysis. SD = standard deviation.

**Table 2 nutrients-14-00491-t002:** Frequency and mean duration of symptoms prior to CD diagnosis in pediatric and adult patients.

Symptoms	Number and Percentage of Patients Reporting Each Symptom				Mean Duration in Years Prior to CD Diagnosis			
	All Patients (*n* = 796)	Children (*n* = 294) ^1^	Adults (*n* = 502) ^1^	*p*	All Patients (*n* = 796)	Children (*n* = 294) ^1^	Adults (*n* = 502) ^1^	*p*
Flatulence	465 (58.4%)	152 (51.7%)	313 (62.4%)	0.003	8.1	3.6	10.3	<0.001
Abdominal pain/Bloating	552 (69.3%)	207 (70.4%)	345 (68.7%)	0.619	7.1	3.1	9.4	<0.001
Chronic fatigue	486 (61.0%)	112 (38.1%)	374 (74.5%)	<0.001	6.3	2.9	7.3	<0.001
Anemia	440 (55.2%)	120 (40.8%)	320 (63.7%)	<0.001	9.2	2.8	11.6	<0.001
Diarrhea	434 (54.5%)	143 (48.6%)	291 (58.0%)	0.011	4.7	2.4	5.8	0.001
Headaches	350 (43.9%)	80 (27.2%)	270 (53.8%)	<0.001	8.6	3.1	10.3	<0.001
Weight loss	337 (42.3%)	118 (40.1%)	219 (43.6%)	<0.001	5.2	3.0	6.4	0.014
Skin rash	289 (36.3%)	105 (35.7%)	184 (36.7%)	0.790	7.4	3.9	9.3	<0.001
Joint pains	269 (33.7%)	60 (20.4%)	209 (41.6%)	<0.001	7.6	3.7	8.8	<0.001
Constipation	271 (34.0%)	83 (28.2%)	188 (37.5%)	0.008	8.5	2.7	11.1	<0.001
Mouth ulcer	256 (32.1%)	58 (19.7%)	198 (39.4%)	<0.001	7.2	2.8	8.5	<0.001
Depression	145 (18.2%)	17 (5.8%)	128 (25.5%)	<0.001	5.8	4.1	6.0	0.298
Ataxia	58 (7.2%)	21 (7.1%)	37 (7.4%)	0.905	5.6	2.9	7.1	0.465
Osteoporosis	53 (6.6%)	10 (3.4%)	43 (8.6%)	0.005	6.5	0.6	7.9	<0.001
No symptoms	26 (3.2%)	13 (4.4%)	13 (2.6%)	0.160	NA ^2^	NA	NA	NA

^1^ The number of children and adults at the age when the diagnosis of CD was made. Chi-squared test was used for statistical analysis. ^2^ NA = not available.

**Table 3 nutrients-14-00491-t003:** Incidence and the mean duration of symptoms after CD diagnosis in pediatric and adult patients.

Symptoms	Number and Percentage of Patients Reporting Each Symptom				Mean Duration in Years after CD Diagnosis			
	All Patients (*n* = 796)	Children (*n* = 294) ^1^	Adults (*n* = 502) ^1^	*p*	All Patients (*n* = 796)	Children (*n* = 294) ^1^	Adults (*n* = 502) ^1^	*p*
Flatulence	333 (41.8%)	93 (31.6%)	240 (47.8%)	<0.001	3.7	2.7	4.1	0.055
Abdominal pain/Bloating	311 (39.0%)	109 (37.0%)	202 (40.2%)	0.377	2.1	2.0	2.2	0.719
Chronic fatigue	278 (34.9%)	57 (19.3%)	221 (44.0%)	<0.001	2.7	2.5	2.8	0.656
Anemia	214 (26.8%)	5 (1.7%)	161 (32.0%)	<0.001	3.0	1.4	3.6	0.001
Diarrhea	203 (25.5%)	54 (18.3%)	149 (29.6%)	<0.001	1.6	1.5	1.6	0.876
Headaches	225 (28.2%)	59 (20.0%)	166 (33.0%)	<0.001	4.7	5.5	4.4	0.363
Weight loss	129 (16.2%)	36 (12.2%)	93 (18.5%)	0.020	2.4	3.0	6.4	0.260
Skin rash	176 (22.1%)	57 (19.3%)	119 (23.7%)	0.157	2.9	2.8	2.9	0.871
Joint pains	181 (22.7%)	43 (14.6%)	138 (27.4%)	<0.001	3.6	4.6	3.3	0.244
Constipation	185 (23.2%)	53 (18.0%)	132 (26.2%)	0.008	2.7	1.8	3.0	0.256
Mouth ulcer	117 (14.6%)	27 (9.1%)	90 (17.9%)	0.001	2.3	2.1	2.4	0.759
Depression	90 (11.3%)	21 (7.1%)	69 (13.7%)	0.005	3.1	3.2	3.1	0.916
Ataxia	32 (4.0%)	11 (3.7%)	21 (4.1%)	0.759	3.0	4.2	2.4	0.336
Osteoporosis	46 (5.7%)	8 (2.7%)	38 (7.5%)	0.005	5.2	1.3	6.0	0.163
No symptoms	155 (19.4%)	80 (27.2%)	75 (14.9%)	<0.001	-	-	-	-

^1^ The number of children and adults at the age when diagnosis was made. Chi-squared test was used for statistical analysis.

**Table 4 nutrients-14-00491-t004:** The prevalence of comorbid diseases among children and adults with diagnosed CD.

Comorbid Diseases	All Respondents (*n* = 796)	Children (*n* = 224)	Adults (*n* = 572)	*p*
Thyroid diseases	23.4%	9.8%	28.8%	<0.001
Depression	12.6%	2.2%	16.7%	<0.001
IgA ^1^ deficiency	10.7%	13.3%	9.6%	0.128
Miscarriages	9.4%	0.0%	13.2%	<0.001
Liver diseases	4.2%	0.9%	5.4%	0.004
Infertility	4.2%	0.0%	5.4%	0.004
Type 1 diabetes mellitus	3.5%	5.3%	2.8%	0.082
Oncological diseases	1.9%	0.0%	2.6%	0.014
Genetic syndrome	1.1%	3.1%	0.4%	0.001
Attention-deficit hyperactivity disorder	1.0%	2.2%	0.5%	0.031
Myocardial infarction	0.6%	0.0%	0.9%	0.159
Stroke	0.4%	0.0%	0.5%	0.276
Other diseases ^2^	40.3%	34.2%	42.7%	0.028
No comorbid disease	16.8%	29.8%	11.7%	<0.001

^1^ IgA—immunoglobulin A. ^2^ Other diseases reported by the respondents were: atopic dermatitis, albinism, bronchial asthma, food and pollen allergy, colitis ulcerosa, Crohn’s disease, endometriosis, epilepsy, ischaemic hearth disease, irritable bowel syndrome, hypertension, gastroesophageal reflux, psoriasis.

**Table 5 nutrients-14-00491-t005:** Diagnostic tools used for CD diagnosis.

	All Respondents (*n* = 796)	Children (*n* = 294) ^1^	Adults (*n* = 502) ^1^	*p*
Serological tests	667 (83.7%)	270 (91.8%)	397 (79.0%)	0.0001
Duodenal biopsy	645 (81.0%)	227 (77.2%)	418 (83.2%)	0.0394
Genetic tests (HLA-typing)	211 (26.5%)	117 (39.7%)	94 (18.7%)	0.0001

^1^ The number of children and adults at the age when CD diagnosis was made. Fisher exact test was used for statistical analysis.

**Table 6 nutrients-14-00491-t006:** Mean number of appointments with GPs ^1^ prior to CD diagnosis about the symptoms, by duration of symptoms.

Duration of Symptoms before the Diagnosis in Years	All Patients (*n* = 796)		Children (*n* = 224)		Adults (*n* = 572)	
	Number of Appointments	Mean	Number of Appointments	Mean	Number of Appointments	Mean
<1	94	7.6	63	9.0	31	4.7
1–5	367	13.5	189	15.9	178	11.0
5–10	164	20.2	31	19.8	133	20.3
10–20	124	28.0	9	20.7	115	28.5
>20	43	36.5	0	-	43	36.5
In total	792 ^2^	17.7	293	15.2	500	19.3

^1^ GPs—General Practitioners. ^2^ Some adults and children did not provide data on the number of appointments with GPs or did not provide data on the duration of symptoms prior to diagnosis. Results are presented as total number of appointments and arithmetic means.

## Data Availability

The datasets generated and/or analyzed during the current study are not publicly available because they are also being used for further ongoing analyses, but are available from the corresponding author on reasonable request.

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
