# Peer review of "Clinical Manifestation and Diagnostic Process of Celiac Disease in Poland—Comparison of Pediatric and Adult Patients in Retrospective Study"

_nutrients, 2022, doi:10.3390/nu14030491_

Round 1
Reviewer 1 Report
Delay in diagnosis of celiac disease is a real problem, so the question of the study is of interest. The paper draws attention to this issue and provide real-life national data about the diagnostic process of celiac disease.
Maior concerns
Point 1. The study is retrospective and questionnaire-based. The data are patient-reported (with high risk of response bias) and not based on medical data, so that data rather reflect patients’ opinion and thoughts. Authors should interpret the results accordingly.
Point 2. The response rate was low, so the representativity of the data is questionable.
Point 3. The correct diagnosis of all celiac patients is also questionable (e.g. the type of the serological test was not determined, the proportion of serology test and biopsy was quite low)
Minor concerns
Point 4. Line 199-200: It is not understandable. What does it belong to?
Point 5. Line 125-126: Table 2: headline: „Mean duration in years prior CD diagnosis” instead of „Mean duration in years after CD diagnosis”?
Point 6. Line 341-343: This sentence is not clear.
Reviewer 2 Report
In this study, the Authors aimed to evaluate the clinical manifestation and diagnostic process of celiac disease (CD) in Polish children and adults.
The members of the Polish Coeliac Society (n=2 500) were asked to complete a questionnaire on socio-demographic factors, clinical and diagnostic aspects of CD. The final analysis was based on 796 responses from patients with confirmed CD diagnosis and included 224 (28.1%) children and 572 (71.9%) adults.
They found that the mean duration of symptoms prior to CD diagnosis in children was significantly shorter than in adults (3.1 and 9 years, respectively). Interestingly, the most frequent symptoms before CD diagnosis were abdominal pain and bloating in children (70.4%), and chronic fatigue in adults (74.5%). Although almost all CD patients claimed to strictly avoid gluten after CD diagnosis, symptoms were still present in the majority of these respondents.
No comorbid diseases were reported by 29.8% of children and by 11.7% of adults (p < 0.001). They concluded that their results indicate that CD diagnosis is delayed in Poland, especially in adults, and clinicians should be aware of the diversity in CD presentation.
The study is of interest as it is well known that clinical presentation is different and geographical differences are reported, and the study focuses on the Poland CD patients.
However, to further improve the manuscript some issues should be addressed.
- It would be specified how CD diagnosis was performed in the study population (pediatric and adult CD patients): international (recognized) vs local (Poland) recommendations. Please, specify serological tests (anti-tissue transglutaminase, anti-endomysial, deamidated gliadin peptides) and histological criteria (Marsh-Oberhuber?) used for the diagnosis.
- A clinically relevant finding is that the duration of symptoms prior to CD diagnosis in children was significantly shorter than in adults (3.1 years and 9 years, respectively). However, the Authors should further discuss the potential impact of delayed diagnosis. In particular, it has been previously reported that a delayed diagnosis may impact secondary autoimmune disorders in CD patients, such as neurological disorders which are frequently marked by the occurrence of some autoantibodies such as anti-neuronal and anti-ganglioside antibodies that should be suggested as part of the work-up of CD patients with neurological manifestations as previously demonstrated (Sera of patients with celiac disease and neurologic disorders evoke a mitochondrial-dependent apoptosis in vitro. Gastroenterology. 2007 Jul;133(1):195-206; Anti-ganglioside antibodies in coeliac disease with neurological disorders. Dig Liver Dis. 2006 Mar;38(3):183-7.). Moreover, a delayed diagnosis and GFD starting may affect the incidence of ostopenia/osteoporosis as recently demonstrated (Celiac Disease Diagnosed through Screening Programs in At-Risk Adults Is Not Associated with Worse Adherence to the Gluten-Free Diet and Might Protect from Osteopenia/Osteoporosis. Nutrients. 2018 Dec 7;10(12):1940.). In light of the above-mentioned literature data, the results of this study are of clinical relevance and would support the need for serological screening for CD in adult patients with atypical clinical presentation (neurological disorders, osteopenia/osteoporosis) and also in atopics (Prevalence of silent coeliac disease in atopics. Dig Liver Dis. 2000 Dec;32(9):775-9).
Author Response
Please see the attachment.
Best regards,
Emilia Majsiak

Round 2
Reviewer 1 Report
I have no further comments